# Feasibility and Error Analysis of Using Fiber Optic Temperature Measurement Device to Evaluate the Electromagnetic Safety of Hot Bridge Wire EEDs

**DOI:** 10.3390/s22093505

**Published:** 2022-05-05

**Authors:** Xuxu Lyu, Guanghui Wei, Xinfu Lu, Haojiang Wan, Xue Du

**Affiliations:** National Key Laboratory of Electromagnetic Environment Effects, Shijiazhuang Campus, Army Engineering University, Shijiazhuang 050003, China; lyuxuxu@163.com (X.L.); luxinfu1988@163.com (X.L.); hbwhj1983@163.com (H.W.); duxue_xdd@163.com (X.D.)

**Keywords:** electromagnetic safety, hot bridge wire EEDs, fiber-optic temperature sensor, temperature measurement accuracy, assessment methodology

## Abstract

Most studies assessing the safety of hot bridge wire EEDs employ temperature sensors that directly use the measurements of the temperature measurement device without analyzing the accuracy of the temperature measurement. This study establishes the response function of the exposed bridge and exposed bridge temperature rise system of hot bridge wire EEDs through the Rosenthal’s temperature rise equation and Laplace transformation as well as experimental tests, and through the response function, the response law and numerical characteristics of the two are compared and analyzed under four typical excitations. Under steady current injection and continuous-wave radiation, both exposed bridge and exposed bridge temperature measurement systems can reach thermal equilibrium, and the equilibrium temperature of both are the same. However, under pulse excitation, the temperature rise measurement value is significantly different from the actual value due to the large difference in response time of the exposed bridge (1 ms) and the exposed bridge temperature measurement device (0.82 s). Studies have shown that under steady current injection and continuous-wave radiation, temperature rise measurements can be directly applied to the safety assessment of hot bridge wire EEDs, while under pulsed conditions, temperature rise measurements cannot be directly applied.

## 1. Introduction

The widespread use of high-power frequency equipment has led to the development of electro-explosion devices (EEDs) which face a very harsh electromagnetic environment [1,2]. The EEDs are employed as ignition components [3,4,5], of which the hot bridge wire EEDs products are most used [6,7]. The hot bridge wire EEDs are relatively sensitive to radio frequency interference (RFI) and electromagnetic pulses (EMP) [8,9], thus their electromagnetic safety must be evaluated to ensure the safety of equipment and systems containing hot bridge wire EEDs.

For assessment of the electromagnetic safety of hot bridge wire EEDs, the inductive testing method represents the traditional method, using a voltage, current sensor, or power meter to measure the bridge wire directly or indirectly. To determine its safety, measurements are obtained in various types of induced current, voltage, or power, the amount of induction, and the experimental hot bridge wire EED safety threshold for comparison [10,11,12]. However, at high frequencies, due to the standing wave effect, the difference in current amplitude between different locations is conspicuous [13], resulting in the use of induction measurement methods being limited in frequency, generally not more than 1 GHz. Researchers have turned their attention to temperature measurement. According to the thermodynamic theory of hot bridge wire EEDs, the electric ignition head uses an electric current to heat the bridge wire, and the bridge wire then heats the surrounding energy-containing agent. When the temperature reaches a certain value, the EEDs explode [14]. That is, whether the hot bridge wire EEDs detonate or not is only related to its bridge wire temperature [15]. The temperature measurement method involves monitoring the bridge wire EEDs electromagnetic safety assessment by monitoring the temperature response of the bridge wire. Fiber-optic temperature sensing devices are commonly used in current experiments of temperature measurement [16,17]. Wang B. et al. [18,19] proposed an equivalent method for testing the electromagnetic margin of EEDs in a strong electromagnetic environment. This method uses thermometry to establish the relationship between the temperature rise and field strength of EEDs in an electromagnetic field, and then obtains the equivalent ignition temperature of EEDs based on their 50% ignition field strength. It then compares this temperature with the equivalent ignition temperature of EEDs to obtain the safety margin. In the same year, the team introduced this method directly into the safety assessment of bridging wires under electromagnetic pulse (EMP) injection. Lu, X. et al. [20] established the relationship between the temperature rise of the exposed bridge wire and the current amplitude at a constant current using the pyrometry method, and obtained the true 50% ignition temperature of EEDs by analyzing the relationship between the heat dissipation coefficients of the exposed bridge wire and the real EEDs, thus providing a test method for the safety of EEDs. Xin, L. et al. [21] employed infrared fiber-optic thermometry for bridge wire EEDs and achieved indirect measurement of their induced currents. However, the above article did not analyze the relationship between the measured temperature rise of the exposed bridge and its true value.

In the literature mentioned above, there were few studies focused on the relationship between the measured temperature rise of the exposed bridge and its true value. Most studies have directly utilized the bridge wire temperature rise measurements under different excitations in the safety assessment of scorched bridge wire type electrical pyrotechnics and have not analyzed the accuracy of the measurements. This paper analyzes and compares the temperature rise response laws of the exposed bridge and exposed bridge temperature measurement systems based on temperature rise theory and system functions. Moreover, this paper discusses what type of excitation caused by the temperature rise can be accurately measured with the temperature measurement device. This is to ensure both the accuracy of the bridge wire temperature measurement data and the effectiveness of EEDs safety assessment results.

## 2. Theory and Calculation

To theoretically analyze and study the temperature rise law of the exposed bridge under various excitations, it is necessary to establish the temperature rise equation of the bridge wire, which is widely used at present in Rosenthal’s temperature rise equation [22]. Rosenthal’s temperature rise equation is a set of total parameter equations in the form of the Equation (1).
(1)$$C_{p}\frac{dT_{c}}{dt}+\gamma T_{c}=P(t)=i^{2}(t)r(T_{c})$$
where i(t) is the current through the bridge wire, $C_{p}$ is the bridge wire heat capacity (J/°C), $T_{c}$ is the bridge wire temperature rise (°C), γ is the bridge wire heat loss coefficient (W/°C^−1^), $r(T_{c})=r_{0}(1+aT_{c})$, $r_{0}$ is the resistance of the bridge wire at room temperature (Ω), a is the bridge wire resistance temperature coefficient (°C^−1^), its value is very small, about 7 × 10^−5^ °C^−1^, and the general pharmaceutical fuming temperature is less than 1000 °C [19], thus change in the resistance of the bridge wire is not considered. The bridge wire resistance is a constant value $r_{0}$, τ is the thermal time constant of the bridge wire (s), and its expression is
(2)$$\tau =C_{p}/\gamma$$

According to Equation (1), incorporating the thermal performance parameters of the bridge wire material can solve the temperature rise response of the bridge wire. Typical hot bridge wire material of hot bridge wire EEDs comprise Ni-Cr material or 6J10, 6J20 material, and the thermal time constant of the bridge wire prepared from the material is usually around 1 ms. Table 1 lists the thermal performance parameters of the bridge wire material of typical hot bridge wire EEDs [23].

### 2.1. Temperature Rise Response Laws of the Exposed Bridge

#### 2.1.1. Steady Current Injection

When the injection current is steady, let $i(t)=I_{0}$, solve Equation (1) to obtain the bridge wire temperature rise as:(3)$$T_{c}(t)=\frac{I_{0}^{2}r_{0}}{\gamma }(1-e^{-\frac{t}{\tau }}).$$

Bringing the parameters of Table 1 into Equation (3), the temperature rise curve of the bridge wire under different amplitudes of steady current injection can be obtained as shown in Figure 1.

It can be seen from Figure 1 that the exposed bridge reaches thermal equilibrium after a certain period under steady current excitation, and it can be seen from Equation (3) that the condition for the bridge wire to reach thermal equilibrium is exp(−t/τ)≈0. Moreover, according to the nature of the exponential function, it is easy to know that the thermal equilibrium condition is satisfied when t=5τ, i.e., when the bridge wire is injected with a steady current, the bridge wire is considered to achieve thermal equilibrium after reaching a 5-fold increase of the thermal time constant of the bridge wire. According to Equation (3), the thermal equilibrium temperature rise of the bridge wire is $I_{0}^{2}r_{0}/\gamma$, that is, from the equilibrium temperature rise of the bridge wire under the injection of steady current $T_{c}\propto I_{0}^{2}$, and from the curve in Figure 1, it can also be observed that the equilibrium temperature rise of the exposed bridge is proportional to the square of the injected steady current amplitude.

#### 2.1.2. Continuous-Wave Radiation

Let the root mean square (RMS) of the induced current of the bridge wire be $I_{0}$, then we have $i(t)=\sqrt{2}I_{0}\sin(\omega t)$, substitute into Equation (1) to solve the equation for the bridge wire temperature rise as:(4)$$T_{c}(t)=\frac{I_{0}^{2}r_{0}}{C_{p}}\cdot \left[\frac{4\omega^{2}\tau (1-e^{-\frac{t}{\tau }})+\tau^{-1}}{\left(4\omega^{2}+\tau^{-2}\right)}-\frac{\tau^{-1}\cos(2\omega t)+2\omega \sin(2\omega t)}{\left(4\omega^{2}+\tau^{-2}\right)}\right]$$

Figure 2 depicts the bridge wire temperature rise curve under sinusoidal current excitation with an RMS value of 100 mA and a frequency of 1500 Hz.

From Figure 2, the fluctuating bridge wire temperature rise can be observed. After reaching about 5-times the bridge wire thermal time constant (5 ms), the bridge wire temperature rise reached a “dynamic equilibrium” state. This state does not mean that the bridge wire reached thermal equilibrium, but that the bridge wire temperature increased at a fixed temperature rise value in the form of sinusoidal fluctuations. The following from Formula (4) in the sinusoidal excitation of the bridge wire temperature rise is employed for analysis.

Let
(5)$$\tau^{-1}\cos(2\omega t)+2\omega \sin(2\omega t)=M\sin(2\omega t+\varphi )$$

Then we have
(6)$$\{\begin{array}{c} M=\sqrt{4\omega^{2}+\tau^{-2}}\\ \begin{array}{c} \varphi =\arctan(\frac{1}{2\omega \tau })\\ \begin{array}{c} \sin\varphi =\frac{\tau^{-1}}{M}\\ \cos\varphi =\frac{2\omega }{M} \end{array} \end{array} \end{array}$$

Bringing Equations (5) and (6) into Equation (4), then we get
(7)$$T_{c}(t)=\frac{I_{0}^{2}r_{0}}{C_{p}}\cdot \left\{\tau [1-(\cos^{2}\varphi )e^{-\frac{t}{\tau }}]-\frac{\sin(2\omega t+\varphi )}{M}\right\}$$

From Equation (7), under sinusoidal current excitation, the bridge wire temperature rise consists of two parts, the first part rises with exponential law, and the second part represents the temperature fluctuation in sinusoidal form. The fluctuating frequency is 2-times the frequency of the injected sinusoidal wave, the expressions of these two parts are as follows:(8)$$\frac{I_{0}^{2}r_{0}}{\gamma }\cdot \left[1-(\cos^{2}\varphi )e^{-\frac{t}{\tau }}\right]$$
(9)$$-\frac{I_{0}^{2}r_{0}}{C_{p}M}\sin(2\omega t+\varphi )$$

From Equation (8), it can be seen that the first part of the bridge wire temperature rise tends to be constant after reaching 5-times the thermal time constant of the bridge wire, with a constant value of $I_{0}^{2}r_{0}/\gamma$. Moreover, from Equation (9) it can be seen that the peak of the temperature rise fluctuation in sinusoidal form is $I_{0}^{2}r_{0}/(C_{p}M)$, and does not change with time, which is in perfect agreement with the trend of the temperature rise curve in Figure 2.

From the analysis of Equation (9), although the sinusoidal form of the temperature rise fluctuation does not vary with time, it decreases rapidly with the increase of the current frequency. Defining the ratio of the peak of the bridge wire temperature rise fluctuation to a constant value $I_{0}^{2}r_{0}/\gamma$ is the relative value of the bridge wire temperature rise pulsation under sinusoidal current excitation, and its expression is:(10)$$\delta_{\sin}=\frac{I_{0}^{2}r_{0}}{C_{p}M}/\frac{I_{0}^{2}r_{0}}{\gamma }=\frac{1}{\tau M}$$

According to Equations (6) and (10), we can find $\delta_{\sin}\in (0,1)$. Figure 3 depicts the variation curve of $\delta_{\sin}$ with the frequency of the sinusoidal current.

From Figure 3, as the frequency of the injected current increases, $\delta_{\sin}$ decreases, and in general, this fluctuation can be ignored when $\delta_{\sin}<1\%$. According to this condition we can obtain:(11)$$\frac{1}{\tau M}<1\%$$

According to the Equation (11) which can be solved to allow the bridge wire temperature rise fluctuation to be neglected, the sinusoidal current should meet:(12)$$f\tau >\frac{50}{2\pi }$$

Considering that the thermal time constant of the material used in the bridge wire of a typical hot bridge wire EED ranges within the millisecond scale, according to Equation (12), f>8 kHz can meet the requirement. The researchers of [24] analyzed the response characteristics of hot bridge wire EEDs in the frequency range from 0–9 GHz based on antenna theory and transmission line theory, and showed that the hot bridge wire EEDs have a sensitive frequency, and its sensitive frequency is closely related to the bridge wire pin size. Moreover, the sensitive frequency of hot bridge wire EEDs can be calculated by Equation (13) [25].
(13)$$f=(2n-1)\frac{150}{L}\mathrm{MHz},n=1,2,3\cdots$$
where L is the length of the hot bridge wire EEDs lead, and the unit is in meters. In general, the length of the EEDs lead will not exceed 1 m, and from Equation (13) it can be deduced that the hot bridge wire EEDs sensitive frequency must be greater than 150 MHz, much greater than the frequency requirements of Equation (12) limited. Thus, in the time period during harmonic electromagnetic field irradiation, the bridge wire temperature rise fluctuation can be ignored as the bridge wire temperature rises only in the first part. According to Formula (6), where at this time φ=0, from comparing the Formulas (3) and (8) the bridge wire temperature rise curve and the amplitude of the sinusoidal current RMS value of the same steady current injection can be observed. That is, the bridge wire can reach thermal equilibrium after 5τ time, with the equilibrium temperature rise of $I_{0}^{2}r_{0}/\gamma$, that is, under sinusoidal excitation, the equilibrium temperature rise of the bridge wire is $T_{c}\propto I_{0}^{2}$.

#### 2.1.3. Single Pulse Injection

When the current injected into the bridge wire is a single-pulse current, let its amplitude be $I_{p}$, the pulse width be $t_{on}$, and the maximum temperature rise of the bridge wire be $T_{c\max}$. Then, the initial conditions corresponding to the differential equation of Equation (1) can be listed:(14)$$\left\{ \begin{array}{l} T_{c}(0)=0,T_{c}(t_{on})=T_{c}{}_{\max},0\leq t<t_{on};\\ T_{c}(t_{on})=T_{c}{}_{\max},T_{c}(\infty )=0,t_{on}\leq t. \end{array}\right.$$

Solve for the bridge wire temperature rise as:(15)$$\left\{ \begin{array}{l} T_{c}(t)=\frac{I_{p}^{2}r_{0}}{\gamma }(1-e^{-\frac{t}{\tau }}),0\leq t<t_{on};\\ T_{c}(t)=\frac{I_{p}^{2}r_{0}}{\gamma }(1-e^{-\frac{t}{\tau }})e^{-\frac{t-t_{on}}{\tau }},t_{on}\leq t. \end{array}\right.$$

From Equation (15), the bridge wire temperature rise is closely related to the pulse amplitude and width of the single-pulse current. Figure 4 depicts the bridge wire temperature rise curve for a single-pulse current injection of 1A with different pulse widths.

It can be seen from Figure 4, when the injection current is a single-pulse current, and the bridge wire temperature rise is within the duration of the maximum temperature rise value, that when the pulse width increases, the bridge filament maximum temperature rise also increases proportionally. From Equation (15), the maximum temperature of the bridge wire is
(16)$$T_{c\max}=\frac{I_{p}^{2}r_{0}}{\gamma }(1-e^{-\frac{t_{on}}{\tau }})$$

The single-pulse current width is very small, generally in the microsecond or even smaller order of magnitude, much smaller than the bridge wire thermal time constant, thus the bridge wire cannot reach thermal equilibrium. According to the infinitesimal equivalence, in a single-pulse injection, the bridge wire maximum temperature rise of $T_{c\max}\propto I_{p}^{2}t_{on}$, and the maximum value of the curve in Figure 4 characteristics can be consistently obtained.

#### 2.1.4. Pulse Train Injection

When the bridge wire excitation is in pulse train form, the effective value of the pulse train is $I_{0}$, the repetition frequency is $f_{r}$, the repetition period is $t_{r}$, the pulse width is $t_{on}$, then the amplitude of the pulse train is $I_{p}=I_{0}/\sqrt{\alpha }$. Where *α* is the duty cycle of the pulse train, that is, the ratio of the pulse width to the pulse repetition period, similar to the solution of the single-pulse injection, then the solution of the pulse train injection under the bridge wire temperature rise in the first cycle can be obtained as:(17)$$\left\{ \begin{array}{l} T_{c}(t)=\frac{I_{p}^{2}r_{0}}{\gamma }(1-e^{-\frac{t}{\tau }}),0\leq t<t_{on};\\ T_{c}(t)=\frac{I_{p}^{2}r_{0}}{\gamma }(1-e^{-\frac{t}{\tau }})e^{-\frac{t-t_{on}}{\tau }},t_{on}\leq t<t_{r}. \end{array}\right.$$

Other cycles can be solved by the iterative method, which is not repeated here.

From Equation (17), the temperature rise of the bridge wire under pulse train injection is related to the amplitude, repetition frequency, and pulse width of the pulse train.

The temperature rise curve of the bridge wire is shown in Figure 5a. When a pulse train current with an amplitude of 1A, a repetition frequency of 15 Hz, 20 Hz, 25 Hz, and a pulse width of 400 ns is injected into the bridge wire; the temperature rise curve of the bridge wire is shown in Figure 5b when a pulse train current with an amplitude of 1A, a repetition frequency of 15 kHz, 20 kHz, 25 kHz, and a pulse width of 400 ns is injected into the bridge wire. The temperature rise curve of bridge wire is shown in Figure 6, where the pulse train injected into the bridge wire has an amplitude of 1A, repetition frequency of 25 Hz, and pulse width of 100 ns, 200 ns, 400 ns pulse train current.

From Figure 5, it can be seen that when the bridge wire excitation is in pulse train form, the different repetition in frequency of the pulse train leads to a significant difference in the temperature response of the bridge wire. Figure 5a shows that at low-frequency pulse train injection, there is a rapid temperature rise of the bridge wire, and one can note that the maximum temperature rise of the bridge wire is independent of the frequency of the injected pulse train. Meanwhile from Figure 5b, it can be seen that the overall temperature rise of the bridge wire at high frequency is exponential, after reaching 5-times the thermal time constant of the bridge wire (5 ms), the temperature rise of the bridge wire reaches a stable interval and fluctuates in this interval. At this time, the maximum and minimum value of the bridge wire temperature rise is fixed, and the maximum and minimum values of the bridge wire temperature rise are $T_{c\max}$ and $T_{c\min}$. Figure 7 depicts the characteristics of the bridge wire temperature rise curve for this phenomenon in detail, in addition, it can be found that as the repetition frequency of the pulse train increases, the bridge wire temperature rise increases and the amplitude of the temperature rise fluctuation decreases.

Figure 6 shows that under low-frequency pulse train injection, the bridge wire temperature rise has reached the maximum at the first pulse width of the pulse, and the maximum temperature rise increases proportionally with the increase of pulse width.

The following is an analysis of the bridge wire temperature rise of the hot bridge wire EEDs under pulse train excitation.

When the bridge wire temperature rise reaches the state shown in Figure 7, the maximum and minimum values of the bridge wire temperature rise can be obtained by taking the initial conditions of the temperature rise in this state into Equation (1) and solving for:(18)$$\left\{ \begin{array}{c} T_{c\max}=\frac{I_{p}^{2}r_{0}}{\gamma }\cdot \frac{1-e^{-\frac{t_{on}}{\tau }}}{1-e^{-\frac{t_{r}}{\tau }}};\\ T_{c\min}=T_{c\max}e^{-\frac{T-t_{on}}{\tau }}. \end{array}\right.$$

Let the equivalence factor be β. The expression is
(19)$$\beta =\frac{1-e^{-\frac{t_{on}}{\tau }}}{1-e^{-\frac{t_{r}}{\tau }}}$$

Then, the maximum value of the bridge wire temperature rise can be expressed as:(20)$$T_{c\max}=\beta \frac{I_{p}^{2}r_{0}}{\gamma }$$

From Equation (20), the maximum value of the bridge wire temperature rise is closely related to the amplitude, repetition period, and duty cycle of the injected pulse, and according to Equations (18)–(20), when $t_{r}\to 0$, according to the infinitesimal equivalence, we get
(21)$$T_{c\max}=T_{c\min}=\frac{I_{0}^{2}r_{0}}{\gamma }$$

As can be seen from Equation (21), when the repetition frequency of the pulse train tends to infinity, the maximum and minimum values of the bridge wire temperature rise tend to a fixed value of $I_{0}^{2}r_{0}/\gamma$. That is, when the repetition frequency of the pulse train injected into the bridge wire is large, the bridge wire temperature rise curve and the injection of a constant current equal to its RMS value are the same.

The relative value of the bridge wire temperature rise pulsation under pulse train excitation is defined as $\delta_{cp}$, and its expression is:(22)$$\delta_{cp}=\frac{T_{c\max}-I_{0}^{2}r_{0}/\gamma }{I_{0}^{2}r_{0}/\gamma }=\frac{\beta -\alpha }{\alpha }$$

Figure 8 depicts the variation of $\delta_{cp}$ with the normalization period $t_{r}/\tau$ and duty cycle α.

From Figure 8, the smaller the $t_{r}/\tau$, the larger the α, and the smaller the $\delta_{cp}$. When $\delta_{cp}<1\%$, the temperature rise pulsation can be ignored, and the temperature rise and steady current injection in the case are the same. Using Equation (22), it is easy to determine that the repetition period and the thermal time constant of the bridge wire at this time should satisfy $t_{r}/\tau \leq 0.04\;(\alpha =0.5)$. At this time, both $t_{r}$ and $t_{on}$ can be considered as infinitesimal of τ, and according to infinitesimal equivalence rewriting Equation (20) can be performed.
(23)$$T_{c\max}=\frac{I_{0}^{2}r_{0}}{\gamma }$$

That is, this case shows $T_{c\max}\propto f_{r}t_{on}I_{p}^{2}$, which is consistent with the pattern shown by the temperature rise curves in Figure 5b and Figure 6.

However, in general, the repetition frequency of the electromagnetic pulse train ranges from tens to hundreds of hertz, and the pulse width is in the range of microseconds or less. According to Equation (22), it can be observed that in the electromagnetic pulse train injection, the relative value of the bridge wire temperature rise pulsation measures much greater than 1%. The temperature rise pulsation cannot be ignored, at this time $t_{on}$ can still be considered τ infinitesimal, and $t_{r}/\tau$ must be greater than five, thus it can be considered that $\exp(-t_{r}/\tau )=0$. From Equation (20), according to the infinitesimal equivalence, the following can be obtained:(24)$$T_{c\max}=t_{on}\frac{I_{p}^{2}r_{0}}{\gamma }$$

From Equation (24), it can be shown that in the electromagnetic pulse train injection, there is a maximum temperature rise of the bridge wire $T_{c\max}\propto I_{p}^{2}t_{on}$, that is, the exposed bridge temperature rise is independent of the repetition frequency. This conclusion and Figure 5a, Figure 6 temperature rise curve laws are the same.

### 2.2. Temperature Rise Response Laws of the Exposed Bridge Temperature Measurement System

The development and maturation of fiber-optic temperature measurement device manufacturing technology has led to its current widespread implementation. Table 2 summarizes the characteristics of several types of current fiber-optic temperature measurement devices.

The temperature rise of the exposed bridge of the hot bridge wire EEDs under excitation is monitored using the most advanced GaAs fiber-optic temperature measurement device (Opsens Radsen-2). The device is used in conjunction with the exposed bridge to form an exposed bridge temperature measurement system. The measurement device uses a GaAs semiconductor wafer as the material of light absorption and reflection with changes in temperature change, independent of time and the environment. It is a truly passive sensor with stable performance, high reliability, fast response time (sampling frequency of 1 kHz), and is unaffected by strong external magnetic fields. The small size of the sensor, with linearity down to the sub-millimeter scale, compared with other types of fiber-optic temperature measurement devices, is especially suitable for applications such as hot bridge wire and other small space temperature measurements. To explore the relationship between the exposed bridge temperature rise measurement results and the actual temperature rise, the response characteristics of the exposed bridge temperature measurement system must be studied. Figure 9 shows the signal flow diagram of the exposed bridge temperature measurement system.

In Figure 9, P(s), $T_{c}(s)$, and $T_{c}^{*}(s)$ are the Laplace transformations of the bridge wire excitation power, the real temperature rise of the bridge wire, and the measured value of the temperature rise, $H_{0}(s)$ and $H_{1}(s)$ are the system transfer functions of the exposed bridge and the exposed bridge temperature measurement device, respectively, and G(s) is the system transfer function of the exposed bridge temperature measurement system.

According to Equation (1), the system transfer function of the exposed bridge is
(25)$$H_{0}(s)=\frac{T_{c}(s)}{P(s)}=\frac{1/\gamma }{\tau s+1}$$

A steady current is injected into the exposed bridge to bring the bridge wire to a unit temperature rise. Figure 10 depicts the measured curve of the temperature measurement device, and the test system diagram is shown in Figure 11.

According to Figure 10, the response time of the temperature measurement system is in the order of seconds, and the response time of the temperature measurement device is recorded as $\tau^{*}$. To accurately measure the response time of the temperature measurement device, the average value is obtained by fitting the experimental data several times, and the fitting results are shown in Table 3.

According to Table 3, the response of the system under steady-state injection can be obtained as:(26)$$T_{c}^{*}(t)=1-e^{-\frac{t}{0.82}}$$

From Equation (26), the response time of the exposed bridge temperature measurement system is 0.82 s, and the Laplace transformation of Equation (26) as:(27)$$T_{c}^{*}(s)=\frac{1}{s}-\frac{1}{s+1/\tau^{*}}$$

Then, the response function of the temperature measurement device is:(28)$$H_{1}(s)=\frac{T_{c}^{*}(t)}{T_{c}(s)}=\frac{\tau s+1}{\tau^{*}s+1}$$

The transfer function of the exposed bridge temperature measurement system is:(29)$$G(s)=H_{0}(s)H_{1}(s)=\frac{1/\gamma }{s\tau^{*}+1}$$

From Equations (29) and (25), the response time of the exposed bridge temperature measurement system changes from τ to $\tau^{*}$, i.e., from milliseconds to seconds compared to the exposed bridge itself. According to Equation (29) and the analysis in Section 2.1, the temperature measurement results of the exposed bridge under different excitations can be analyzed theoretically. That is, based on the exposed bridge temperature rise results under different excitations analyzed in Section 2.1, the following analysis is obtained using Laplace inversion $T_{c}^{*}(t)={ℒ}^{-1}[ℒ[P(t)]G(s)]$:Steady current excitation;
(30)$$T_{c}^{*}(t)=\frac{I_{0}^{2}r_{0}}{\gamma }(1-e^{-\frac{t}{\tau^{*}}})$$

When the probe reaches thermal equilibrium, the measured value has no error with the theoretical value and satisfies:(31)$$T_{c}^{*}\propto I_{0}^{2}$$

That is, the bare bridge temperature rise measurement is proportional to the square of the amplitude of the steady-state current.

2.Continuous-wave excitation;

When the excitation source is continuous-wave irradiation, according to Section 2.1.2, only the first part of the temperature rise of the bridge wire is considered, then we have
(32)$$T_{c}^{*}(t)=\frac{I_{0}^{2}r_{0}}{\gamma }(1-e^{-\frac{t}{\tau^{*}}})$$

When the probe is in thermal equilibrium, the measured value has no error with the true value and satisfies:(33)$$T_{c}^{*}\propto I_{0}^{2}$$

That is, the exposed bridge temperature rise measurement is proportional to the RMS value of the sine wave.

3.Single-pulse excitation;

When the excitation is a single pulse, then:(34)$$T_{c}^{*}(t)=\frac{I_{p}^{2}r_{0}}{\gamma }(1-e^{-\frac{t}{\tau^{*}}})(\varepsilon (0)-\varepsilon (t_{on}))$$

From Equation (34), it can be shown that there is an error between the measured value and the true value under this excitation, and since the pulse width of a single pulse is below the microsecond order, the maximum temperature rise can be obtained according to the infinitesimal equivalence to satisfy:(35)$$T_{c\max}^{*}\propto I_{p}^{2}t_{on}$$

That is, the exposed bridge temperature rise measurement is proportional to the squared amplitude and pulse width of a single pulse.

4.Electromagnetic pulse string excitation.

When the excitation is an electromagnetic pulse train, then:(36)$$T_{c}^{*}(t)=\frac{I_{p}^{2}r_{0}}{\gamma }\left[\varepsilon (0)-\varepsilon (t_{on})\right]\frac{n(e^{\frac{t_{r}}{\tau^{*}}}-1)+e^{-\frac{t}{\tau^{*}}}(1-e^{\frac{nt_{r}}{\tau^{*}}})}{e^{\frac{t_{r}}{\tau^{*}}}-1}$$
where *n* is the number of repetition periods of the pulse train, because the repetition period of the electromagnetic pulse train is generally in the range from tens to hundreds of hertz. Thus, it can be regarded as the infinitesimal of Formula (36) after the infinitesimal equivalent replacement to obtain:(37)$$T_{c}^{*}(t)=\frac{I_{p}^{2}r_{0}}{\gamma }floor(\frac{t}{t_{r}})(\varepsilon (0)-\varepsilon (t_{on}))$$
where floor(x) is to the left of the integer function, according to Formula (37) one can obtain the excitation of the exposed bridge temperature rise measurement value, the true value of the error, and the maximum temperature rise to meet:(38)$$T_{c\max}^{*}=\frac{t_{on}}{t_{r}}\cdot \frac{I_{p}^{2}r_{0}}{\gamma }$$

That is, to meet:(39)$$T_{c\max}^{*}\propto t_{on}f_{r}I_{p}^{2}$$

That is, the measured temperature rise of the exposed bridge is proportional to the squared amplitude, the repetition frequency, and the pulse width of the electromagnetic pulse train.

### 2.3. Summary and Comparison

In summary, the analysis summarizes the response law and characteristics of bare bridge and bare bridge temperature measurement systems under typical excitation as shown in Table 4.

## 3. Testing and Results

To illustrate the correctness of the theoretical analysis, we measured the temperature rise of the bridge wire under the above four typical excitations using an exposed bridge temperature measurement system, the test results are shown below.

## 4. Discussion

According to Figure 12, the measured temperature rise of the bridge wire is proportional to the square of the current amplitude under steady and constant current injection; under continuous-wave radiation, the bridge wire temperature rise measurement value is proportional to the square of the field strength; under single-pulse injection, the measured temperature rise of the bridge wire is proportional to the square of the pulse amplitude and the pulse width; under pulse train injection, the measured temperature rise of the bridge wire is proportional to the square of the pulse train amplitude, the pulse width, and the repetition frequency. The experimental results are in high agreement with the theoretical analysis, thus demonstrating the correctness of the theoretical analysis.

## 5. Conclusions

In this paper, the feasibility of evaluating the safety of electromagnetic radiation of hot bridge wire EEDs using fiber-optic temperature measurement devices is analyzed from the perspective of error analysis. This is based on Rosenthal’s temperature rise equation and Laplace transform. The following conclusions can be drawn from the theoretical and experimental studies:Under steady current injection or continuous-wave radiation, the bridge wire can reach thermal equilibrium, and its thermal equilibrium temperature rise is proportional to the steady current amplitude or continuous-wave RMS squared. Under single-pulse excitation, the bridge wire will not reach thermal equilibrium, and the maximum temperature rise of the bridge wire is proportional to the squared amplitude and the pulse width of the single pulse. Under pulse train excitation, the bridge wire will also not reach thermal equilibrium, and its maximum temperature rise is proportional to the squared amplitude and the pulse width of the pulse train, independent of the repetition frequency of the pulse train.The response of the exposed bridge temperature measurement system was tested by a steady current injection test. Moreover, the Laplace transform was performed on the excitation and response of the exposed bridge as well as the exposed bridge temperature measurement system to determine the system transfer function of both, i.e., Equations (25) and (29). By comparing the system transfer function of both, it can be shown that both the exposed bridge and the exposed bridge temperature measurement system are first-order inertia links. However, the time constants of both differ significantly, with the time constant of the exposed bridge being in the millisecond range and the time constant of the exposed bridge temperature measurement system being in the order of seconds, subsequently leading to a significant difference in the temperature rise response law of both.The temperature rise measurements of the exposed bridge under different excitations were analyzed by the system transfer function of the exposed bridge temperature measurement system. The analysis showed that the response laws of the exposed bridge and the exposed bridge temperature measurement system were consistent under steady current injection, continuous-wave radiation, and single-pulse injection. Under pulse string injection, the response laws of both were not consistent, and the temperature rise measurements of the exposed bridge temperature measurement system were not only proportional to the square of the pulse train amplitude and the pulse width, but also proportional to the repetition frequency of the pulse train.Only under constant current injection and continuous-wave radiation, can the exposed bridge reach thermal equilibrium. The measured value of the exposed bridge temperature rise is essentially equal to the actual value, and the measured value of the temperature rise can be directly applied to the safety assessment of the hot bridge wire EEDs. Under pulse excitation, the exposed bridge cannot reach thermal equilibrium and the response time of the exposed bridge temperature measurement system is excessively long. Therefore, the measured value is different from the actual value, and the measured value of the temperature rise cannot be directly applied to the safety assessment of hot bridge wire EEDs.

However, this paper does not analyze how the temperature rise of the bare bridge can be accurately measured under pulsed excitation. This represents one of the future research points in solving the problem of inaccurate temperature measurement owing to the long response time of the temperature measurement device under pulse excitation, by correcting the measured value of the bare bridge temperature measurement system.

## Figures and Tables

**Figure 1 sensors-22-03505-f001:**
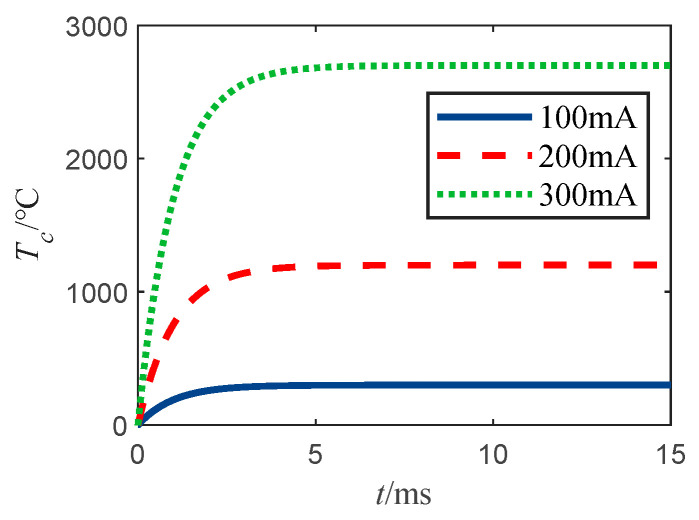
Temperature rise curve of bridge wire under different amplitudes of steady current injection.

**Figure 2 sensors-22-03505-f002:**
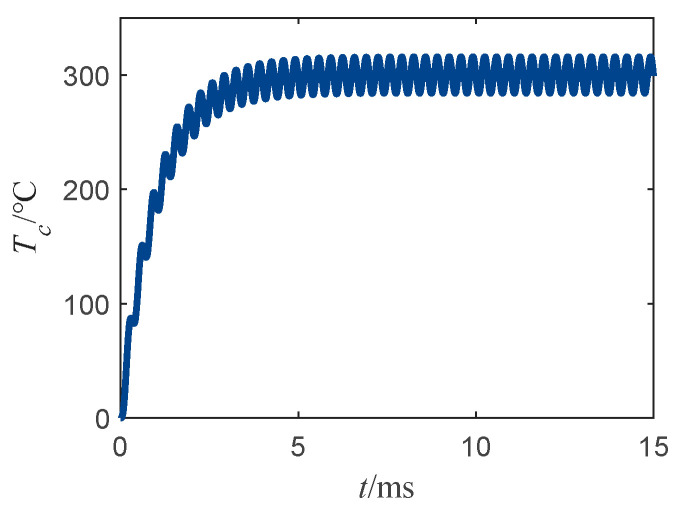
Temperature rise curve of bridge wire under sinusoidal current excitation.

**Figure 3 sensors-22-03505-f003:**
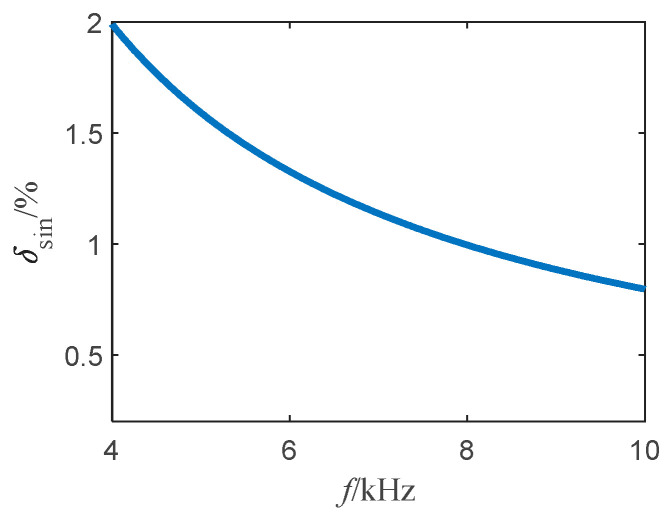
The curve of $\delta_{\sin}$ with the frequency of sinusoidal current.

**Figure 4 sensors-22-03505-f004:**
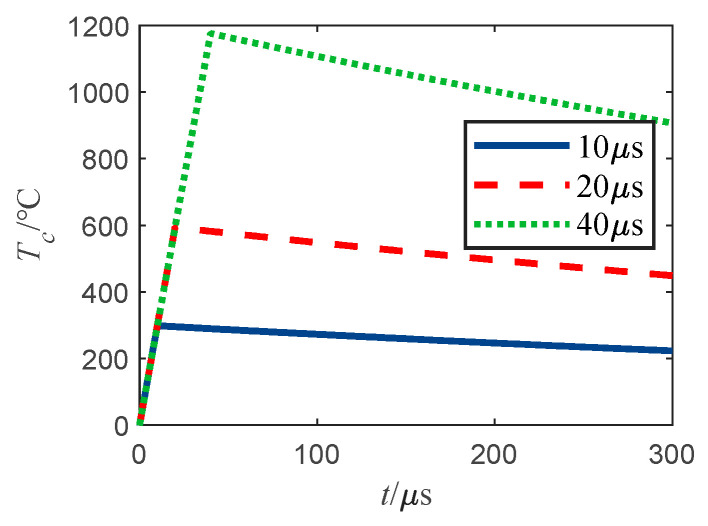
Bridge wire temperature rise curve under single-pulse current injection.

**Figure 5 sensors-22-03505-f005:**
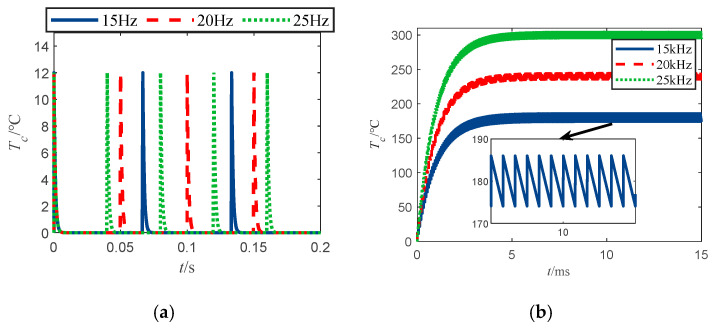
Temperature rise curve of bridge wire under different frequency pulse train injections. (**a**) Bridge wire temperature rise for low-frequency pulse train excitation. (**b**) Bridge wire temperature rise for high-frequency pulse train excitation.

**Figure 6 sensors-22-03505-f006:**
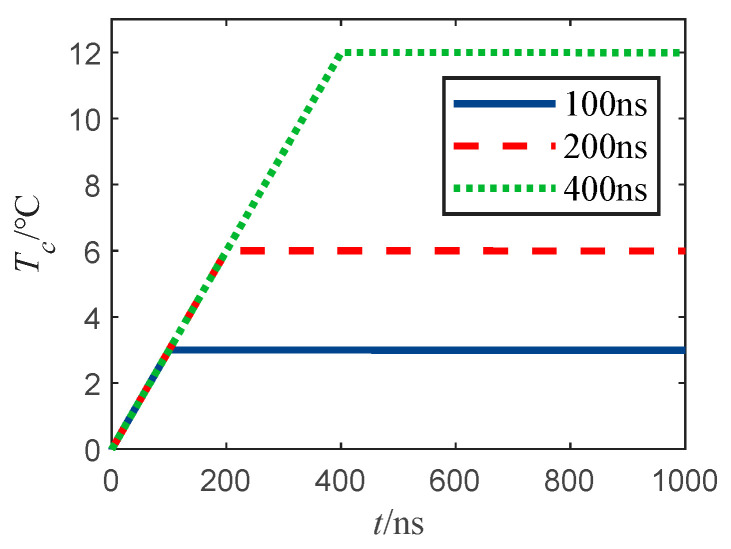
Temperature rise curve of bridge wire under pulse train injection with different pulse widths.

**Figure 7 sensors-22-03505-f007:**
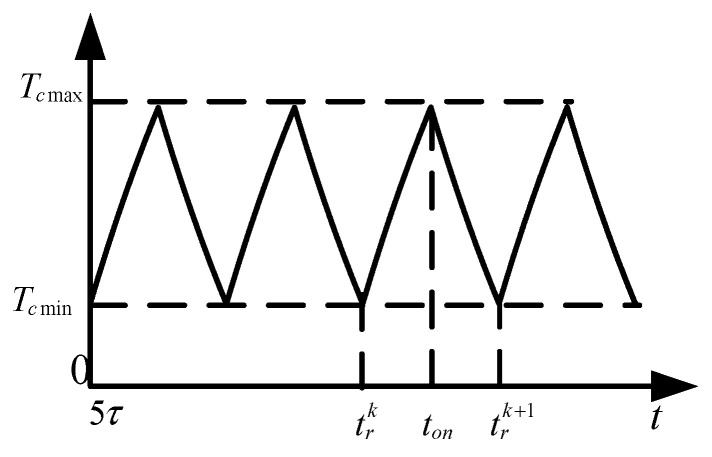
Maximum and minimum temperature rise diagram.

**Figure 8 sensors-22-03505-f008:**
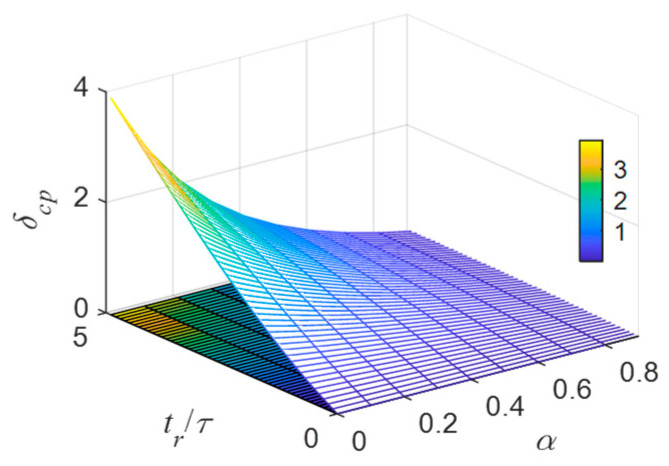
The variation law of $\delta_{cp}$ with $t_{r}/\tau$ and α.

**Figure 9 sensors-22-03505-f009:**
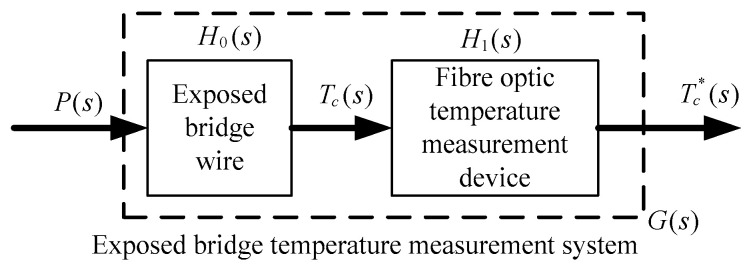
Signal flow diagram of exposed bridge temperature measurement system.

**Figure 10 sensors-22-03505-f010:**
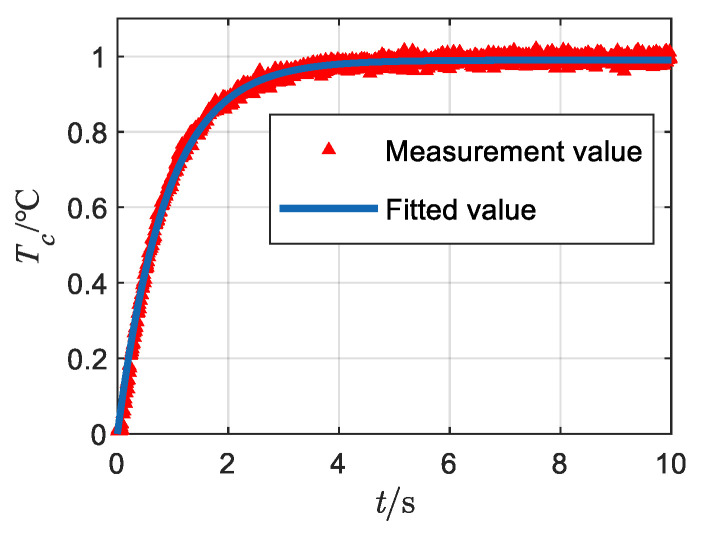
Fiber-optic temperature measurement system response curve.

**Figure 11 sensors-22-03505-f011:**
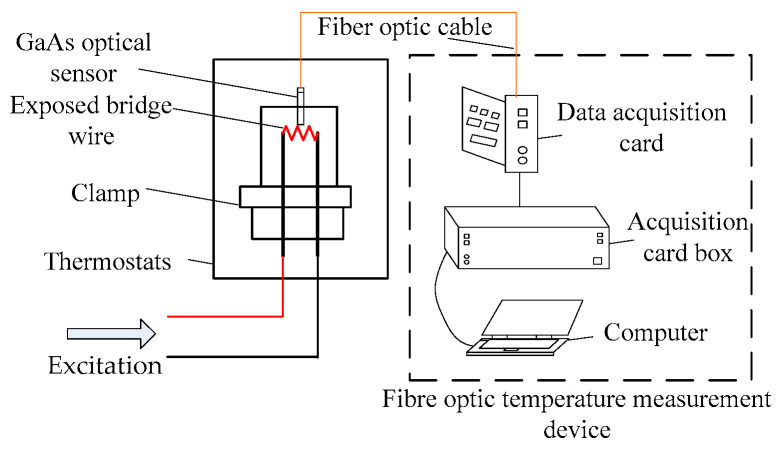
Exposed bridge temperature measurement system.

**Figure 12 sensors-22-03505-f012:**
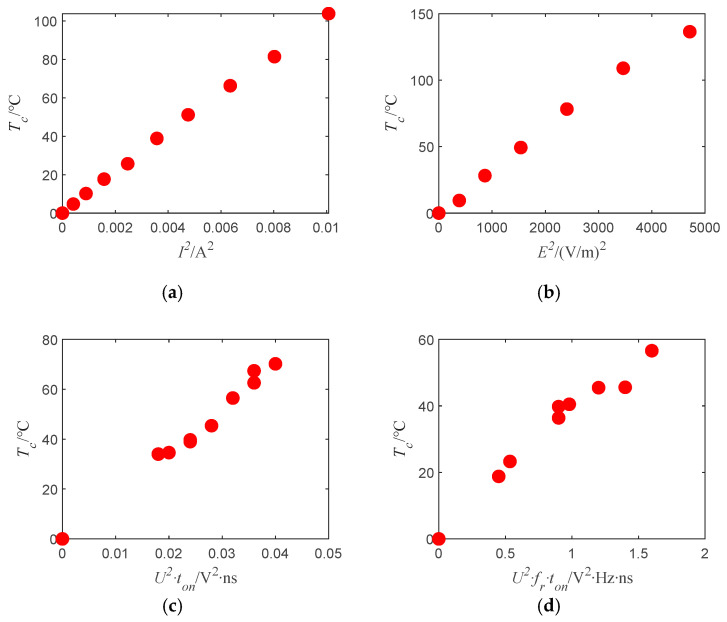
Results of exposed bridge temperature rise test: (**a**) Temperature rise measurement results under steady current injection; (**b**) The relationship between field strength and temperature rise measurement results under continuous-wave irradiation of 300 MHz; (**c**) Bridge wire temperature rise measurement results for single-pulse parameter changes; (**d**) Bridge wire temperature rise measurement results when the pulse train parameters change.

**Table 1 sensors-22-03505-t001:** Thermal property parameters of bridge wire.

Parameters	Value
*C_P_*/μJ·°C^−1^	0.2
*γ*/mW·°C^−^^1^	0.2
*τ*/ms	1
*r*_0_/Ω	6

**Table 2 sensors-22-03505-t002:** Characteristics of several types of fiber-optic temperature measurement devices. Adapted from Ref. [17]. 2020, Mikolajek, M.

Type	Measurement Error	Response Time
Fiber infrared	Larger	Smaller
Fiber optic fluorescent	Large	Large
White light interference	Small	Large
GaAs	Small	Small

**Table 3 sensors-22-03505-t003:** Fitting results of the measured values of the exposed bridge temperature measurement system.

Serial No.	Target Function: *a*(1 − exp(−*t*/*τ**))		R-Squared
	*a*	*τ**	
#1	0.9987	0.8373	0.9991
#2	1.0141	0.8176	0.9997
#3	0.9984	0.8321	0.9989
#4	1.0112	0.8088	0.9995
Average	1	0.82	0.9993

**Table 4 sensors-22-03505-t004:** Comparison of temperature rise response of exposed bridge and exposed bridge temperature measurement system.

Type	Exposed Bridge	Exposed Bridge Temperature Measurement System	Value Comparison
Steady state	*T_c_* ∝ $I_{0}^{2}$	$T_{c}^{*}$ ∝ $I_{0}^{2}$	Equivalent
Sine continuous wave	*T_c_* ∝ $I_{0}^{2}$	$T_{c}^{*}$ ∝ $I_{0}^{2}$	Equivalent
Single pulse	*T_c_*_max_ ∝ *t_on_*$I_{p}^{2}$	$T_{c\max}^{*}$ ∝ *t_on_*$I_{p}^{2}$	Unequal
Pulse train	*T_c_*_max_ ∝ *t_on_*$I_{p}^{2}$	$T_{c\max}^{*}$ ∝ *t_on_f_r_*$I_{p}^{2}$	Unequal

## Data Availability

The authors confirm that the data and materials supporting the findings of this study are available within the article.

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
