# Peer review of "Feasibility and Error Analysis of Using Fiber Optic Temperature Measurement Device to Evaluate the Electromagnetic Safety of Hot Bridge Wire EEDs"

_sensors, 2022, doi:10.3390/s22093505_

Round 1

Reviewer 1 Report

This work has merit and addresses an important topic. However,  the following items need to be addressed:

Comment 1:

pg. 10, lines 316-317
The authors state " the most advanced GaAs fiber optic temperature measurement device and the exposed bridge to form a exposed bridge temperature measurement system"

It is recommended the authors thoroughly describe the GaAs fiber optic used.  Is the fiber optic commercially available or developed internally? Beyond the sampling rate, what are the key features or considerations for selecting the fiber optic? What is the outer diameter and length of fiber?  

Comment 2:  1pp. 7-8 
Figures 5 and 6 should have higher resolution on the y-axis.  For example, every 50 units for Figure 5b instead of every 100 units.

Reviewer 2 Report

Th

The authors of the manuscript” Feasibility and error analysis of using fiber optic temperature measurement device to evaluate the electromagnetic safety of hot bridge wire EEDs” are proposed and experimentally realized the response function of the exposed bridge and exposed bridge temperature rise system of hot bridge wire electro-explosive devices through the Rosenthal's temperature rise equation and Laplace transformation. Some information must be addressed before publication.

  1. Although it will not affect much for understanding, in the overall manuscript some typos need to be corrected or clarified before printing for publication (e.g. section 1: Introduction last paragragh, line 73 Check the punctuation error).
  2. Can the authors add a reference here: the general pharmaceutical fuming temperature is less than 1000 ℃, so do not consider the change in the resistance of the bridge wire.
  3. Can the authors explain: the bridge wire temperature rise reached a "dynamic equilibrium" state, this state does not mean that the bridge wire reached thermal equilibrium.
  4. Check Figure 11 to follow the uniform font and text size over all figures in the manuscript.
  5. It is appreciable if the authors tabulate the existing temperature measurement devices and the proposed scheme.
  6. The authors said “this novel scheme can approach the temperature rise measurement value … the large difference in response time of the exposed bridge (1 ms) and the exposed bridge temperature measurement device (0.82 s).” According to the Eq. (26) in manuscript, the response time of the exposed bridge temperature measurement system is too long. Is it or not? If it is, is there any better way to solve this? If it is not, why?

It is a useful guide for the temperature measurement to evaluate the electromagnetic safety of hot bridge wire electro-explosive devices. This manuscript can be accepted for publication with some revisions.

Reviewer 3 Report

Manuscript No:  Sensors-1684004

Title:  Feasibility and Error Analysis of Using Fiber Optic Temperature Measurement Device to Evaluate the Electromagnetic Safety of Hot Bridge Wire EEDs

Authors:  Xuxu Lyu, Guanghui Wei, Xinfu Lu, Haojiang Wan and Xue Du

  1. Overview
  2. In this manuscript the authors report on work on analysis of feasibility and error analysis of using fiber optic temperature measurement device to evaluate the electromagnetic safety of hot bridge wire EEDs.
  3. The contents are expressed clearly; the manuscript is well organized and it is written in reasonable English.
  4. The authors have acknowledged recent related research.
  5. As long as my knowledge, the work presented is original and it is correct from a scientific point of view.

  1. Detailed analysis

Abstract: Be clear, objective. State briefly what you did, how did you do it, the quantitative results you and the novelty of your work. Please make it as synthetic as you can. Please organize the ideas in each paragraph.

  1. Introduction: provides an interesting approach to the subject and there are up to date references.

  1. Overall assessment

In my opinion the work can be published after minor correction in the text

  1. Review Criteria
  2. Scope of Journal

Rating: Medium

  1. Novelty and Impact

Rating: Medium

  1. Technical Content

Rating: Medium

  1. Presentation Quality

Rating: Medium
